# *Cyprinid herpesvirus 3* Evolves In Vitro through an Assemblage of Haplotypes that Alternatively Become Dominant or Under-Represented

**DOI:** 10.3390/v11080754

**Published:** 2019-08-15

**Authors:** Sandro Klafack, Anna-Sophie Fiston-Lavier, Sven M. Bergmann, Saliha Hammoumi, Lars Schröder, Walter Fuchs, Angela Lusiastuti, Pei-Yu Lee, Sarahi Vega Heredia, Anne-Sophie Gosselin-Grenet, Jean-Christophe Avarre

**Affiliations:** 1Institute of Infectology, Friedrich-Loeffer-Institut, Federal Research Institute for Animal Health, 17493 Greifswald-Insel Riems, Germany; 2ISEM, IRD, CNRS, EPHE, University of Montpellier, 34095 Montpellier, France; 3Institute of Molecular Virology and Cell Biology, Friedrich Loeffer Institut, Federal Research Institute for Animal Health, 17493 Greifswald-Insel Riems, Germany; 4Research Institute for Freshwater Aquaculture and Fisheries Extension, Bogor 16129, Indonesia; 5GenReach Biotechnology, Taichung City 407, Taiwan; 6Faculty of Sciences, University of Montpellier, 34095 Montpellier, France; 7DGIMI, University of Montpellier, INRA, 34095 Montpellier, France

**Keywords:** Cyprinid herpesvirus 3, cell culture passages, virus evolution, haplotype sequence

## Abstract

Viruses are able to evolve in vitro by mutations after serial passages in cell cultures, which can lead to either a loss, or an increase, of virulence. Cyprinid herpesvirus 3 (CyHV-3), a 295-kb double-stranded DNA virus, is the etiological agent of the koi herpesvirus disease (KHVD). To assess the influence of serial passages, an isolate of CyHV-3 (KHV-T) was passaged 99 times onto common carp brain (CCB) cells, and virus virulence was evaluated during passages through the experimental infections of common carp. After 78 CCB passages, the isolate was much less virulent than the original form. A comparative genomic analysis of these three forms of KHV-T (P0, P78 and P99) revealed a limited number of variations. The largest one was a deletion of 1363 bp in the predicted ORF150, which was detected in P78, but not in P99. This unexpected finding was confirmed by conventional PCR and digital PCR. The results presented here primarily suggest that, CyHV-3 evolves, at least in vitro, through an assemblage of haplotypes that alternatively become dominant or under-represented.

## 1. Introduction

Common carp (*Cyprinus carpio L.*) is the most produced fish in the world. With its ornamental breed, the koi is also one of the most expensive fish. It is highly susceptible to the *Cyprinid herpesvirus 3* (CyHV-3), also called Koi Herpesvirus (KHV). Since its first report in the late 1990s [1], KHV disease (KHVD) has spread to many countries worldwide [2], and is recognized as a significant problem for the common carp and koi aquaculture industries [3,4]. Together with carp pox virus (*Cyprinid herpesvirus 1*), goldfish herpesvirus (*Cyprinid herpesvirus 2*), and eel herpesvirus (*Anguillid herpesvirus 1*), CyHV-3 clusters as a member of the genus *Cyprinivirus* in the family, *Alloherpesviridae* [5]. It causes mass mortality in carp aquaculture, often with more than 80% losses and severe symptoms. Most common symptoms are gill necrosis, skin haemorrhage, excessive mucus production, later on sandpaper skin, and/or sunken eyes [1,6]. Once infected with KHV, carps bear the virus for life and will, therefore, act as carriers [7,8,9]. Moreover, persistent infected carp can shed the virus for the rest of their life. Then, those carp or koi can infect naive fish and spread the virus.

Several genomes of CyHV-3 have now been sequenced [10,11,12,13]. With 156 open reading frames (ORF) and a length of 295 kb, KHV has the biggest genome known in herpes viruses [14]. Unfortunately, little is known about the function of the 156 ORFs or their importance in the virulence. Because of this knowledge gap, it is difficult to target a particular ORF for vaccine production [15,16]. Thus, classical methods, like their attenuation along cell culture passages [17], were used to generate vaccine strains. Although virus attenuation, by successive passages on cell culture, has proven efficient [18], no genomic and/or transcriptomic comparisons, between original and attenuated viral strains, have been carried out so far, in view of understanding the molecular mechanisms of virus attenuation. It was recently shown that mutations occurring in the genome of CyHV-3, during cell culture passages, could be reverted after additional passages [19]. The occurrence of multiple haplotypes, within the same viral specimen, is now well documented [12,20,21], and these genetic reversions could be the result of a switching dominance of one, or several, haplotypes over the others. The present study aimed at testing this hypothesis, by making phenotypic and genomic comparisons of a single virus isolate, before passage on cell cultures (P0), and after 78 passages (P78) and 99 passages (P99). The results presented below prove, for the first time, that CyHV-3 evolves, at least in vitro, through an assemblage of haplotypes that alternatively become dominant and under-represented.

## 2. Materials and Methods

### 2.1. CyHV-3 Propagation Onto CCB Cells and Virus Harvest and Storage

Common carp brain (CCB) cells [22] were grown at 20 °C in minimal essential medium with Earls’ salts (Invitrogen), supplemented with 10% FBS, 10mM HEPES, 2.2 g/L NaHCO (Roth), 1% non-essential amino acids (Biochrom), and 0.12 g/L pyruvic acid sodium salt (Merck). Twenty four-hour-old CCB cell monolayers were adsorbed for 1 h at 20 °C with an isolate of CyHV-3, collected from an infected koi in Taiwan (KHV-T) [19]. Culture medium was then added to the KHV-T inoculated cells, which were subsequently incubated at 20 °C. Viral suspensions were collected after the consecutive passages when 90% of cells showed cytopathic effects (CPE), usually after 7 days. For virus harvest, the flasks were frozen at −80 °C and thawed once at room temperature. One hundred µL of virus suspension was used directly for new inoculation and adsorption of a new flask (25 cm²). The remaining suspensions were stored at −80 °C. KHV-T was serially passaged over 100 times.

### 2.2. Experimental Infections of Carp

Previous results had shown that genetic mutations frequently occurred between passages 78 and 99 [19]. For this reason, the wild-type KHV-T (P0) and the resulting viruses after 78 (P78) and 99 (P99) passages were examined for their ability to elicit both, mortality and morbidity in common carp. For this purpose, six groups of forty six-month old carp were placed in 250-L aquaria (total = 240). Each group corresponded to one treatment and kept in an independent aquarium. Prior to challenge, fish were adapted to the re-circulating systems for 14 days at 20 °C (+/−1 °C). First, samples consisting of gill swab and blood for serum preparation were collected before infection. Carp were then immersed for 1 h in 10-L tanks, containing the respective virus at a dose of 10^5^ TCID_50_/mL for P0 and P99, and at doses of 10^4^, 10^5^, and 10^6^ TCID_50_/mL for P78. Fish from the control group were similarly immersed in 10-L tanks containing no virus. After that, they were transferred back to their original aquaria. Gill swabs and sera were collected every 14 days from five fish per aquarium. Mortality and morbidity were recorded daily by counting dead fish and carp with symptoms, mainly increased mucus production, and local necrosis on the skin and the gill, respectively. Dead fish were immediately examined for the presence of KHV by PCR, using the protocol described in Bergmann et al. [23]. All experiments on animals proceeded under the strict German rules for animal welfare and were legally authorized by the animal protection commission of Mecklenburg, Western Pomerania, with the allowance no. 7221.3-1-008/18 (TV 02/18 KHV-att 4). As the commission did not allow to use more fish for this experiment, it was not possible to make experimental replicates.

### 2.3. Extraction of Viral DNA, Library Preparation, and Genome Sequencing of P78 and P99

Genomic DNA was extracted from both the cell pellets and cell supernatants, using the Nucleospin virus extraction kit (Macherey Nagel, Düren, Germany). Frozen virus cultures were thawed on ice, aliquoted in volumes of 450 µL, and centrifuged at 3000 g for 1 min. Supernatants were transferred into new tubes and mixed with 400 µL of lysis buffer supplemented with 10 µL of proteinase K provided in the kit, whereas cell pellets were re-suspended in 400 µL of lysis buffer, that was supplemented with 10 µL of proteinase K. Viral lysis was achieved by a 15-min incubation at 70 °C. DNA was then purified following the manufacturer’s instructions, and eluted in 30 µL of PCR-grade H_2_O. The purity of the obtained DNA sequences was checked by spectrophotometry (Nanodrop 2100), while their quantity was measured using a Qubit (ThermoFisher Scientific, Waltham, MA, USA).

DNA libraries were prepared with the Nextera XT DNA Library Prep kit (Illumina, San Diego, CA, USA), using 1 ng of input DNA. DNA tagmentation, amplification, and purification were carried out exactly as specified by the provider. The average size distribution of DNA fragments was verified with a BioAnalyzer 2100 (Agilent Technologies Santa Clara, CA, USA), and the concentration of each DNA library was measured by qPCR, according to Illumina’s recommendations. Six replicate DNA libraries of P78 and P99 were prepared by the IMHE Master students, in the frame of a practical training session. Among these technical replicates, two from P78 cell pellets (P78-1c and P78-2c), two from P78 cell supernatants (P78-1s and P78-2s), and two from P99 cell pellets (P99-1c and P99-2c), chosen according to their comparable final concentration (comprised between 11.3 and 33.7 nM), were sequenced at GenSeq platform (Montpellier, France) on a single lane of a MiSeq instrument with MiSeq Reagent Kit v2 (Illumina, San Diego, CA, USA), using 500 cycles (2 × 250 bases, paired-ends). 

### 2.4. Preparation and Genome Sequencing of KHV-T (P0)

The wild-type form of KHV-T had already been sequenced in a previous project [12] without being released, and was used here for genome comparisons. After extraction with the Wizard Genomic DNA purification kit (Promega), DNA was subjected to a specific target enrichment, as described by Hammoumi et al. [12], and sequenced on a HiSeq2000 platform (Illumina, San Diego, CA, USA) together with 7 other specimens in a paired-end (2 × 100 bases) format, at GenSeq platform (Montpellier, France). 

### 2.5. Genomic Sequence Analysis

The seven sequence datasets (P78-1c, P78-2c, P78-1s, P78-2s, P99-1c, P99-2c, and P0) were first analyzed by BKD Masters students, with the help of some authors of this article, in the frame of a practical training session. The quality of the raw sequencing data was evaluated using fastQC (version 0.11.8; https://github.com/s-andrews/FastQC). Potential adaptor sequences were removed with Trimmomatic (version 0.35) [24], and bases with a quality score > 30 were selected using SeqTk (version 1.3-r106; -q 0.001; https://github.com/lh3/seqtk). The reads were then mapped to the KHV-J strain reference genome (accession number AP008984) with BWA-mem (version bwa V0.7.9a) [25], using the default parameters. Reads aligning with a quality lower than 30 (except those aligning twice in the terminal repeats of the reference genome) were filtered out. Single nucleotide polymorphisms (SNPs) and short indels (<100 bp) were named, combining Picard (version 1.61) and GATK (version 3.5), with a minimum quality filter at Q20. The effects of each variant were then assessed using the “bedtools intersect” (v2.27.0) and “SnpEff” (v4.3) tools. For both tools, the annotation version 1 from the reference genome (AP008984), downloaded from https://www.ncbi.nlm.nih.gov/nuccore/AP008984.1/, was used. For large variant calling (>100 bp), read depth along the reference genome was first computed using Samtools with the command “depth” (version 1.9) [26], and then compared between P78, P99, and P0. The detection of large variations was confirmed by visualization of the read mapping with the Integrative Genomics Viewer (IGV, V2.4.7) [27], and manual curation allowed the precise determination of their boundaries. The regions of interest were finally annotated using KHV-J reference genome (annotation release 1), and putative protein motifs, within predicted open reading frames (ORFs), were searched with the NCBI Conserved Domains (CD) search tool (https://www.ncbi.nlm.nih.gov/cdd/) [28]. Raw sequences (fastq files) were stored in the public Sequence Read Archive (SRA) repository and can be accessed under the accession PRJNA511566. 

### 2.6. PCR Assays

Two PCR assays that target the regions, nt 258055–260647 (encompassing the whole ORF150) and nt 258428-258927 (inside the deletion), were carried out using two primer pairs, designed from P0 sequence: ORF150_-363_F / ORF150_+343_R and ORF150_11_F / ORF150_520_R (Table 1). DNA extracted from P0, P78, and P99 was amplified using the GoTaq G2 kit (Promega) under the following conditions: 95 °C for 5 min, 30 cycles of 95 °C for 30 s, 60 °C for 30 s, and 75 °C for 3 min, followed by a final extension at 75 °C for 5 min. PCR products were run onto agarose gels, visualized with ethidium bromide under a UV trans-illuminator and recovered from the gel for further Sanger sequencing.

### 2.7. Digital PCR Assays

For digital PCR, two primer-probe sets were designed: One located inside the deletion (nt 131-243 of ORF150) and one located downstream (nt 1548–1694 of ORF150, Table 1). DNAs extracted from P0, P78, and P99 were diluted 1:1000 in H_2_O, and both diluted and undiluted DNAs were amplified in triplicate, using the Naica system (Stilla Technologies, Villejuif, France). This system allows the PCR reaction mix to be divided into a high number (usually between 20,000 to 30,000) of very small droplets (or partitions), each containing a small number of target molecules. PCR amplification measurements in each partition (here via a fluorescent probe that binds to the amplified DNA) gives access to the number of target molecules contained in the initial sample (https://www.gene-pi.com/). A unique PCR solution, containing 1× of PerfeCTa qPCR ToughMix (QuantaBio, Beverly, MA, USA), 40 nM fluorescein (QuantaBio, Beverly, MA, USA), 250 nM of each probe and 200 nM of each primer was prepared. One µL of DNA was added to 24 µL of this solution and loaded onto the wells of the Sapphire chips. Partitioning was performed in the Naica Geode, according to Stilla’s instructions, and the amplification programme consisted of an initial denaturation at 95 °C for 10 min, followed by 45 cycles of 95 °C for 30 s, and 60 °C for 40 s. The chips were transferred to the Naica Prism instrument and partitions, read using 3 fluorescent channels: Blue for probe ORF150_195_P, green for fluorescein (background fluorescence), and red for probe ORF150_1589_P (Table 1). 

Crystal droplets were counted with the Crystal Miner software (Stilla Technologies, Villejuif, France), and concentrations of targeted nucleic acids were automatically calculated using the default fluorescent thresholds. Diluting each DNA to 1:1000 enabled the uncertainty, related to the concentration calculation, reduce below 3% in the red channel. For the undiluted P78, the uncertainty in the blue channel varied between 4.7 and 5.3%. Two-dimensional graphs, showing red versus blue fluorescence intensities, were generated with all the droplets from the 3 replicates of each DNA sample. DNA concentrations were expressed as copy/µL ± SD and were compared between the red channel (region 1548–1694 of ORF150), which reflects the concentration of both haplotypes, and the blue channel (region 131–243 of ORF150), which reflects the concentration of the undeleted haplotype only. All primers and probes were synthesized by Eurofins, and all qPCR and dPCR experiments were run at the qPHD platform of Montpellier University.

## 3. Results and Discussion

In order to evaluate whether a natural attenuated live vaccine can be obtained, an isolate of CyHV-3, collected from an infected koi in Taiwan (KHV-T) was serially passaged on CCB cells over 100 times at 20 °C, and selected passages were tested for attenuation with common carp. Among the selected passages, P78 and P99 exhibited the most interesting features. Firstly, P78 elicited lower mortality and morbidity rates than the P0 (wild-type) or P99 viruses (Figure 1). Indeed, the mortality induced by P78, during an experimental infection, dropped from 90% (P0) to 18% with an infection dose of 10^5^ TCID_50_/mL and to 0 with 10^4^ TCID_50_/mL infection dose (Figure 1A,B). Likewise, symptoms were completely absent in the P78 group at 10^5^ TCID_50_/mL, while the morbidity induced by P0, at the same dose, reached 90% and was mostly associated with severe symptoms (Figure 1D,E), consistent with what is usually observed for naturally occurring CyHV-3 isolates [29]. The mortality induced by P99 (28%) was higher than that obtained with P78. Likewise, morbidity was also much higher than that recorded for P78, with a peak of 100% at 9 days post-infection; however, the symptoms caused by P99 did not last long and essentially consisted of increased mucus (Figure 1F). Even though, in the absence of replicates, it was not possible to infer whether these differences are significant, it seems that P99 partially recovered in its virulence ability.

The sequencing of four P78 and two P99 technical replicates led to a number of reads between 1297956 and 3315745, resulting in a mean depth of coverage, ranging between 262× and 568× (Table 2). These values were much higher for P0 (60049308 reads and 7802× of mean depth coverage) because it had been subjected to specific target enrichment prior to sequencing on a HiSeq platform. KHV-J was used as mapping reference because KHV-T had previously been shown to belong to the Asian lineage [19]. As one could expect, the number of variations identified at the genome scale between the three forms of KHV-T and KHV-J was very low (Table 2). Variations were conserved among replicates, and the slight discrepancies, observed between the four P78 replicates (46–54, Table 2), mostly concerned tandem repeated regions. Comparing P78 and P99 with P0 revealed a limited number of variations, which increased with the passages (Table 2. The majority of these variations consisted of single nucleotide substitutions or short insertions/deletions of repeated motifs, within variable numbers of tandem repeats, without any influence on the reading frame. Only three of these mutations elicited a frameshift in a predicted ORF of P78 (Appendix A) and of P99 (Appendix A). With a few exceptions, most of these variations had a frequency lower than 1, indicating the presence of more than one variant. The only large variation (>100 bp), observed between the 3 KHV-T forms, corresponded to a 1363-bp deletion, located in KHVJ160 (corresponding to ORF150) of P78. This deletion was observed in the 4 technical replicates of P78. Most notably, the read depth, comprised between ~200x and ~450x, at positions 258153 and 259517 (depending on the replicate), suddenly dropped to 0x between these positions, indicating that no reads were recovered from this region during sequencing in none of the four replicates. Most surprisingly, this deletion was totally absent in the two replicates of P99, as indicated by an even read depth upstream, along and downstream of the 1363-bp deleted region (Figure 2). Although, it is acknowledged that the rate of read mis-assignments in multiplexed sequencing runs may account for 0.06 to 0.29% of the total number of reads [30,31,32], This cannot explain the complete absence of reads in the deleted region of P78. Since the possibility of a ‘reverse mutation’ of the same length, and at the same location, can be ruled out, it is likely that P78 also contained the undeleted variant (ORF150). The presence of mixed genotypes has been demonstrated to be a common feature of CyHV-3 infections in vivo [12,20,21], but also in vitro [19]. One can therefore assume that the haplotype, carrying the full ORF150, was present in P78, but in too low proportions for being detected through the sequencing and/or analysis protocol we used. Errors accumulated during every step of a sequencing protocol, including library preparation, sequencing, and read alignment. For illustration, it was shown that a variant present at a frequency of 2.5% in the initial sample will not be accurately detected at a read depth lower than 1000x [33,34]. 

To verify these sequencing results, a conventional PCR assay was designed to target the full ORF150, and its deleted region was applied on the three KHV-T samples. Using the first set of primers targeting the full ORF150, a unique band was observed around 1200 bp for P78. Whereas for P0 and P99, the PCR product had the expected length around 2593 bp (Figure 3). Additionally, the second set of primers did not yield any amplification in P78, while a 500-bp fragment was amplified in P0 and P99, as was expected with this primer set (Figure 3). Sanger sequencing of all the obtained amplicons showed 100% identity with P0 sequence. These PCR results clearly confirmed that this deletion did not result from a sequencing artefact. Although often overlooked, the extent of information available in a sample is limited by the quantity of biological material in the sample prior to its amplification [35]. Moreover, it is also known that amplification by PCR may distort the content of a sample through different sources of errors, such as efficiency biases, stochasticity, template switches, or polymerase errors [36]. These errors may have a significant impact on sequence representation, especially for sequences that are present at very low copy numbers [36,37]. As reflected by the weak percent of mapped reads (around 30%), the amount of viral sequences in the input DNA (ratio of CyHV-3 / host reads) was relatively low. Taking all these elements together, these results seemed to indicate that the deleted variant was very abundant in P78, but under the detection threshold in P99. In other words, the main haplotype of P78 carried the deletion of 1363 bp, whereas the dominant haplotype in P99 was the same as in P0, the wild-type KHV-T.

To confirm these findings, a quantitative digital PCR was carried out with two primer-probe sets, designed inside, and outside, the deleted portion of ORF150 (Table 1, Figure 4A). The assay should lead to two amplicons (one in each channel) if the deletion is absent, and to only one amplicon (in the red channel) in the presence of the deletion (Figure 4B). When applied to P78 DNA, a small proportion of partitions were amplified in the blue channel, leading to an estimated concentration of the undeleted haplotype of 162 ± 7 copies/µL (Figure 4C). However, since all partitions were positive in the red channel, preventing any reliable concentration measurement, a 1:1000 dilution was necessary to estimate a concentration of 298 ± 51 copies/µL (Figure 4D). This resulted in a ratio of 5.43E-4 (162/(298*1000)), between the undeleted haplotype and all haplotypes. This means that the undeleted haplotype represented only 0.054% of the viral population, and unambiguously confirms that the undeleted variant was present in P78 in an extremely low proportion, and explains why it could not be detected by either, Illumina sequencing or regular PCR. The deletion was also searched in P0 and P78, using the same assay (Figure 4E,F). The results revealed a nearly equal concentration of target DNA in both channels, indicating that the deleted haplotype was not detected in these samples. The analysis of intermediate passages (from P10 to P90) revealed that the deleted variant started to be detected at passage 50 (at a proportion of 30%), and could not be detected any longer after passage 90 (not shown). If we cannot rule out that the deleted variant was under the limit of quantification [38], these results demonstrate that the haplotype composition may change in remarkable proportions within several cell culture passages. 

The 1363-bp deletion starts between genes KHVJ159 (ORF149) and KHVJ160 (ORF150), and spans more than half of ORF150 (see Figure 2). As a result, the original initiation codon is missing, and an alternative ORF150 could potentially begin at position 259570, providing this resulting truncated ORF is transcribed. As already identified by Aoki et al. [10], and confirmed by a new search against the conserved domain database, ORF150 contains a RING-HC (Really Interesting New Gene) finger domain in its N-terminal region (e-value = 1.23 × 10^−7^; https://www.ncbi.nlm.nih.gov/protein/129560669). RING fingers, and specifically, HC (C3HC4) types, can bind two zinc cations and are involved in ubiquitination, a potent regulator of cellular protein function, such as oncogenesis, viral replication, or apoptosis [39,40,41]. As this RING motif is missing in the truncated form of ORF150, and since this deletion is the major difference between P78 and P0, it is tempting to associate this absence to the drop of virulence, or partial attenuation, observed for P78. Preliminary results on the deletion of the same 1363-bp fragment, and of the entire ORF150 in the original KHV-T isolate, indicated a clear attenuation of virulence [42]. If these findings need to be confirmed, the fact that P99 did not recover the level of virulence, that were observed for P0, does not support this potential genotype-phenotype association. P99 contains 30 additional mutations, compared to P0 that were not present in P78 (Table 2). Some of these mutations may also contribute to virus attenuation, which could explain such a partial recovery. Further studies need to be carried out, in order to evaluate the actual role of ORF150 in CyHV-3 virulence, on the one hand, and the role of these additional mutations, on the other hand.

To conclude, our results demonstrate that an in vitro infection by CyHV-3 is the result of a mixture of haplotypes that co-occur, and that the ratio between these haplotypes may dramatically vary along infection cycles. To study what drives the haplotype composition during an infection will require the use of other computational and/or sequencing approaches, as for instance, those developed for haplotype reconstruction in RNA viruses [43,44,45].

## Figures and Tables

**Figure 1 viruses-11-00754-f001:**
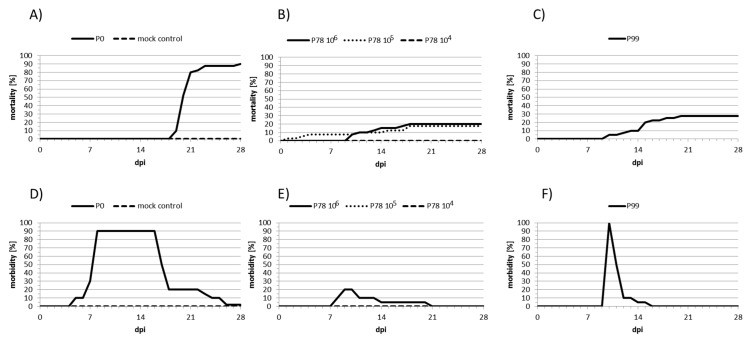
Mortality and morbidity, elicited by the three forms of KHV-T. Carp were experimentally infected with P0 and P99 at a dose of 10^5^ TCID_50_/mL, and with P78 at doses of 10^4^ TCID_50_/mL, 10^5^ TCID_50_/mL and 10^6^ TCID_50_/mL. Mortality (**A**–**C**) and morbidity (**D**–**F**) were recorded during 28 days post-infection (dpi) and plotted at scale.

**Figure 2 viruses-11-00754-f002:**
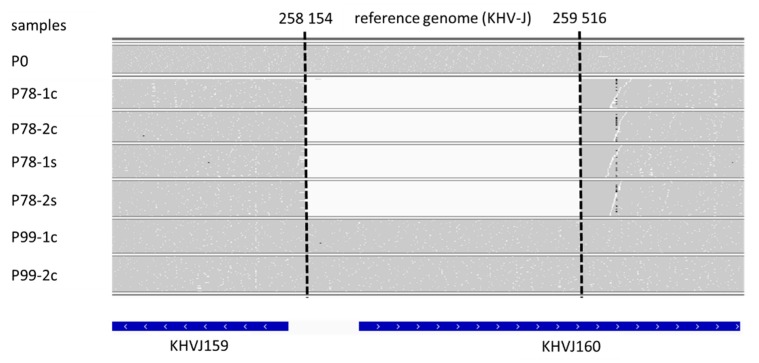
Integrative Genomics Viewer (IGV) screenshot of the region containing the large deletion for the seven sequenced samples. Each horizontal track corresponds to the reads mapped on the KHV-J reference genome (AP008984). White areas correspond to an absence of reads and dashed lines indicate the boundaries of the deletion. Gene annotation of KHV-J is shown at the bottom. KHVJ159 corresponds to ORF149 and KHVJ160 to ORF150, according to Aoki et al. [10].

**Figure 3 viruses-11-00754-f003:**
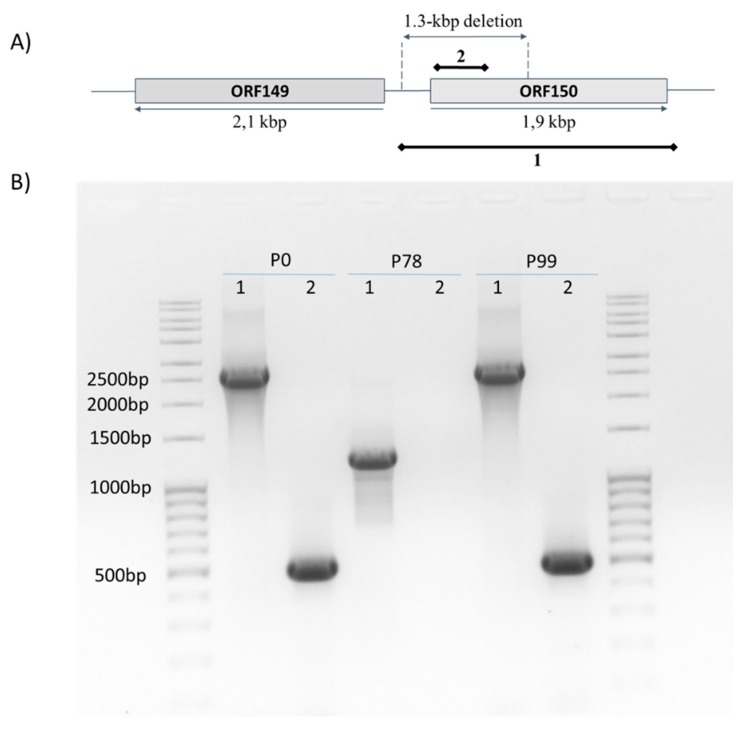
PCR results using the primer set covering the whole ORF150 (1) and the primer set designed inside the deletion (2). (**A**) Schematic representation of the assay design. The expected amplicon sizes according to P0 are 2593 bp, and 500 bp using the two primer sets, respectively. (**B**) The two external lanes were loaded with a 100-bp molecular weight marker, and the corresponding sizes are indicated on the left. 1 and 2 refer to the primer pairs ORF150_-363_F / ORF150_+343_F and ORF150_11_F / ORF150_510_R, respectively (see Table 1).

**Figure 4 viruses-11-00754-f004:**
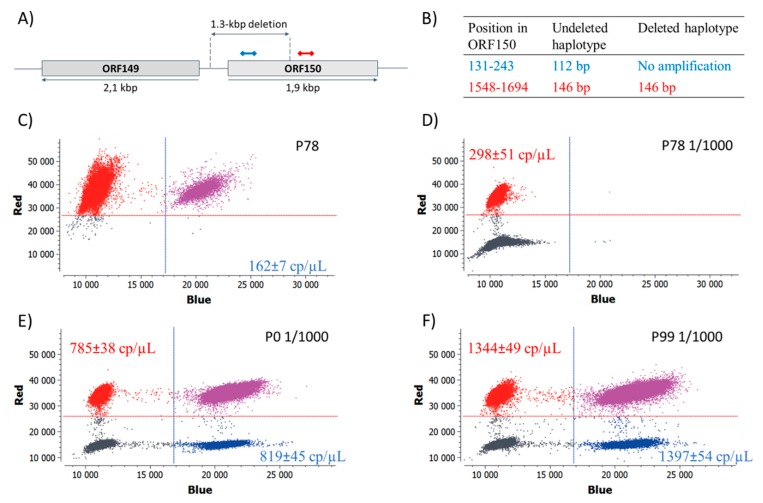
Quantification of deleted and undeleted haplotypes by digital PCR. (**A**) Schematic representation of the assay design and, (**B**) expected amplification products. (**C**–**F**) Fluorescence intensities of all the partitions in the red and blue channels, and estimated concentrations of the corresponding amplicons (mean ± SD of 3 replicates, in copy/µL). The red channel reflects the concentration of both haplotypes, and the blue channel reflects the concentration of the undeleted haplotype only. Purple dots represent the partitions that are positive in both channels, and the grey dots correspond to negative partitions. Y and X axes represent the fluorescence intensity in the red, and blue channels, respectively.

**Table 1 viruses-11-00754-t001:** List of primers and probes used for conventional and digital PCR.

Primer/Probe ^a^	Sequence	5’ Position ^b^
Conventional PCR		
ORF150_-363_F	GCGTCGACGGAGCATG	258055
ORF150_+343_R	CGAAAGAGTAAGCCGTTGCC	260647
ORF150_11_F	CACAAGAGATGGACGCTCAG	258428
ORF150_510_R	GTTCTCGCCCAGCACCA	258927
Digital PCR		
ORF150_131_F	GCTGGACCTGTCACAATTCTAT	258548
ORF150_195_P (FAM/BHQ1)	TCGCACCGTCGTCAAGCAGT	258612
ORF150_243_R	TGGTCCAGTAGACGGTTGA	258659
ORF150_1548_F	GAGCGAGGAACTCTACACAAC	259965
ORF150_1589_P (Cy5/BHQ1)	TGAGGATGCAGAAGCAGTGGATGT	260006
ORF150_1694_R	GGTAAGGGTAAAGCAGACCATC	260110

^a^ Numbering refers to the first nucleotide of ORF150 (“−” means upstream and “+” downstream); ^b^ According to KHV-J sequence.

**Table 2 viruses-11-00754-t002:** Main features of genome comparisons.

Sample	Number of Reads ^a^	% Mapped Reads	Mean Coverage [1st-3rd Quartile]	Number of Variants Against KHV-J ^b^	Number of Variants Against P0 ^b^
P0	60,049,308	98.62	7802 [7868–7932]	80	-
P78-1c	1,297,956	37.86	262 [202–292]	46	21
P78-2c	2,179,255	37.07	418 [301–466]	54	26
P78-1s	3,002,298	27.08	356 [248–398]	50	25
P78-2s	3,315,745	32.28	568 [448–634]	49	23
P99-1c	3,280,716	30.30	559 [404–633]	103	58
P99-2c	2,227,886	31.94	395 [302–451]	103	57

^a^ All reads with a quality < Q30 were removed; ^b^ Only variations < 100 bp were considered here.

## Data Availability

Raw sequences (fastq files) were stored in the public Sequence Read Archive (SRA) repository and can be accessed under the accession PRJNA511566.

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
