# Peer review of "Cyprinid herpesvirus 3 Evolves In Vitro through an Assemblage of Haplotypes that Alternatively Become Dominant or Under-Represented"

_viruses, 2019, doi:10.3390/v11080754_

Round 1

Reviewer 1 Report

The study entitled “Cyprinid herpesvirus 3 evolves in vitro through an assemblage of haplotypes that alternatively become dominant or under-represented” by S. Klafack and colleagues describes the generation of a CyHV-3 viral strain that is attenuated after 78 passages in carp brain cells. As a result of the serial passage, the virus is now much less virulent, inducing low mortality and morbidity in 6-month old naïve carp. Genomic analysis of the virus at passages 0, 78 and 99, and show that at passage 78 the most prevalent variant in the viral quasispecies has a 1363pb deletion. However in passage 99, the dominant CyHV-3 variant lacks the large deletion and can induce moderate mortality and sharp morbidity in infected animals. The work concludes that CyHV-3 can evolve by drifting into a new quasispecies makeup where one variant is strongly dominant.

The experiment is well designed and executed; however there are no experimental replicates, which undermines the work’s reach. The manuscript reads very well, in proper written English. The figures are in general poorly designed and need revising.

Comments:

The sequencing data should be equivalent for all sequenced samples in coverage and depth. This does not seem to be the case (lines 211-13). Why was P0 subjected to specific target enrichment and the other isolates not?

Table 1 is not provided in the manuscript. Do lines 216, 217 and 219 refer to supplementary table 1? (Table S1).

Figure 1 looks fine; however, the text that refers to this figure needs to be revised. Line 196 should read “figure 1A and 1B”, since P0 and P78 (105) mortality are being compared in this sentence (a drop from 90% to 18%). Likewise, line 198 should read “figure 1D and 1E”, since P0 and P78 morbidity is being compared in this sentence.

Two things stand out as unclear in figure 2. First, several of the elements above the 7 “tracks” are undefined: the long, thin box at the very top, with a number (APxxxxx which can’t be read but is probably the reference NCBI sequence), the small red rectangle, the doube-pointed arrow with a number, and the additional long thin box below that, with new numbering. All these element pertain in one way or another to the viral genome, either in entirety or fragments, the analyzed areas, etc. Second, there is one long dark box/thick line above the two viral ORFs, which is unclear what it is. Either label or remove it. In conclusion, please include only the important elements needed to understand this figure, and explain what those elements are. Also, please define IGV at the beginning of the figure legend.

Figure 3 must be improved. As it is, the reader has to guess which amplicon in the gel comes from which virus and the primers used. At minimum, each lane in the gel [including the molecular weight markers] should be clearly labeled with either the sample name or simple numbering 1 to 6 that must then be explained in the figure legend. Molecular sizes should be indicated to the left or right of the gel. A simple diagram similar to the one presented in figure 4A could be included here as panel 3A, showing the position of the different primers used. Likewise, the legend to figure 3 is incomplete, and numbers like 1 and 2 refer to something that is nowhere in the figure.

Digital PCR is a fairly new concept and probably not known to most readers. A minimum effort to briefly outline the principle must be made, not just the methodology (like the primers, temperature or concentration of probe). It can’t be simply outlined in the methods section as the “Neica system”. Addition of something along the lines of “...This is a system that divides/distributes the target nucleic acid into very small aliquots (or partitions), allowing for a more accurate measurement of PCR amplification in each partition via a fluorescent probe that binds to the amplified DNA”. In this way the word ‘partition’ is now understood. In addition, authors should specify the fluorescence type/color that each probe is conjugated to (lines 168-169) so that when mentioning the channels, it is understood which amplicon you are referring to.

In line with the above paragraph, the process by which the authors reach the numbers described in the paragraph on page 8 (lines 275-291) is not clear (for example 162+/- 7 or 298+/- 51 copies/ul, or the number “0.054% of the viral population”). The analysis must be outlined either in the methods or in the results section. While it is true that the calculations may be aided by the software, the reader must be able to follow the reasoning. Due to the lack of explanation on how the system works (see previous paragraph) phrases like “the red channel”, “both channels” or “proportion of partitions” lack proper meaning. Please revise this section to explain how you reach your conclusions of low target sequence concentration in the samples.

Line 277 should read “(one in each channel)” rather than “(in both channels)”

Figure 4 needs units in the axes. Are these relative fluoresence units (RFUs)?

The discussion is short and focuses only on the speculation about the deletion of the RING-HC sequences in P78. This speculation related to the importance of this RING-HC motif for virulence may not be relevant because, as the authors point out, the P99 dominant variant does contain the RING motif but did not recover the virulence. The authors could delete the 1363bp from the P0 in a future study to test this hypothesis.

Importantly, due to the lack of experimental replication in this study there is no way of knowing whether or not in a future experiment the dominant variant could be a completely different virus from p78, or it could be identical to wt (or P0). Therefore, the authors should discuss whether similar studies have found that replications of this experiment would lead to obtaining the same variants or not, to bring some light to this study.

The most interesting finding is that the virus population or quasispecies can drift with the serial passages, which can lead to the generation of a particularly interesting viral variant. However, whether this variant is stable over time after being generated is neither addressed in this study nor discussed. Are there similar studies that give any hints about the stability of the generated variants? This is an important discussion point given that attenuation of CyHV-1 could lead to the generation of a vaccine.

p { margin-bottom: 0.1in; direction: ltr; font-variant: normal; color: rgb(0, 0, 0); letter-spacing: normal; line-height: 115%; text-align: left; break-inside: auto; background: transparent none repeat scroll 0% 0%; text-decoration: none; break-before: auto; break-after: auto; }p.western { font-family: "Arial", serif; font-size: 11pt; font-style: normal; font-weight: normal; }p.cjk { font-family: "Noto Sans CJK SC Regular"; font-size: 11pt; font-style: normal; font-weight: normal; }p.ctl { font-family: "FreeSans"; font-size: 12pt; font-style: normal; font-weight: normal; }

Reviewer 2 Report

Koi herpesvirus (KHV) is an alphaherpesvirus that causes mortality in carp aquaculture. Vaccine strains of KHV have been developed by serial passage of the virus in cell culture leading to highly attenuated viruses. Little is known with regards to the genetic changes that result in virus attenuation. In the present studies, the authors set out to determine the genetic and phenotypic composition of KHV that was serial passaged in cell culture.  Following serial passages of the virus, they tested the P0, P78 and P99 isolates for pathogenicity and sequenced the entire genome for each of the three passage isolates.  Both the P78 and P99 viruses were severely attenuated compared to the P0 virus.  Sequence analysis identified multiple nucleotide mutations (primarily single base changes) and a few small deletions. The one major difference found for the P78 virus was a 1363 bp deletion that was not found in the P99 virus. The deletion had the potential to alter the expression of three KHV genes. What’s surprising is that the P99 virus did not contain the large deletion which is confusing since the authors demonstrate that all of the P78 (99.95%) contained the deletion.  So that raises the question of how does the virus deletion get repaired if the sequence is absent from the P78 isolates.  This is especially puzzling since this change was found in just an additional 20 passages (see comments below).  Overall, the studies are rather limited in scope, simply confirming that multiple serial passages can lead to virus attenuation.  Since there are numerous mutations it would add significance if the importance of some of these could sorted out.

Specific points

1.  To determine the importance of the 1363 bp deletion with regards to attenuation the deletion should be introduced into the wild type KHV strain (P0 virus) by either marker rescue or using KHV BAC.  The multiple mutations that the P78 contains make it impossible to determine if the deletion is important for attenuation.

2.  To determine at what point during the passaging the deletion starts to show up the PCR studies should be done with several early P viruses (P10, 20, 30, 40, etc.) in order to determine when and how prevalent this deletion is in the virus.  The early isolates are available so why not do the simple PCR studies.

3.  As mentioned the repair of the deletion in the P99 stocks is hard to explain.  If the deletion did get repaired between passage 78 and 99 than there should be a gradual increase in the repaired virus between those 20 passages with the repaired virus eventually outcompeting the deletion virus. Again, this should be examined by PCR analysis for several of the P78 to P99 viruses.

4.  Fig. 3.  Labels should be added to indicate what samples are in each lane. Size markers need to be added next to the two outside lanes.

Round 2

Reviewer 2 Report

The authors have addressed the major issues from the previous reviews and the studies are in line with the scope and type of papers that appear in Viruses.  With that said, the authors indicated that they have additional data that will be included in a future paper and some of that data appears to be relevant to addressing some of the points raised in my initial review.  They should at least refer to that data in the current paper.  For example, regarding point 1, the discussion should include reference to the fact that the 1363 bp deletion alone results in attenuation.  Also, (points 2 and 3) since I still find it surprising that the deletion is lost within 12 passages if the data is available it should be noted what proportion of the viruses in the passaged (P78-P90) stocks contain the deletion.  Just would like an indication of how fast the deletion virus is replaced.

1. To determine the importance of the 1363 bp deletion with regards to attenuation the deletion should be introduced into the wild type KHV strain (P0 virus) by either marker rescue or using KHV BAC. The multiple mutations that the P78 contains make it impossible to determine if the deletion is important for attenuation.

We have performed homologous recombinations to delete the same fragment (1363 bp) andalso the entire ORF150. Both led to the same results: attenuation. However, since the main focus of this manuscript is the change of haplotype composition during in vitro passages, these results are not described here. They have been included in a dedicated manuscript thathas been submitted recently.

2. To determine at what point during the passaging the deletion starts to show up the PCR studies should be done with several early P viruses (P10, 20, 30, 40, etc.) in order todetermine when and how prevalent this deletion is in the virus. The early isolates are available so why not do the simple PCR studies.

Conventional PCR is not sensitive enough to evaluate the abundance of viral haplotypes, as shown in the manuscript. This is why digital PCR was applied. As suggested by reviewer, we collected several passages that were stored in the freezer and analyzed them. The deleted variant started to be detected at passage 50 (at a proportion of 0.3), and after a peak at passage 78, could not be detected any longer after passage 90. This will all be detailed in a coming manuscript, together with the results described below (point 3).

3. As mentioned the repair of the deletion in the P99 stocks is hard to explain. If the deletion did get repaired between passage 78 and 99 than there should be a gradual increase in the repaired virus between those 20 passages with the repaired virus eventually outcompeting the deletion virus. Again, this should be examined by PCR analysis for several of the P78 to P99 viruses.

The deletion is not getting repaired: these observed variations are the result of different virus subpopulations that alternatively dominate along cell passages, as shown in the manuscript. According to our knowledge on the replication strategy of CyHV-3, we are now able to influence the composition of the variants that are in the population. Our latest results indicate that the same virus and virus titers, replicated at 15, 20 and 26°C, can lead to different dominant subpopulations within 10 passages. This was made with at least three CyHV-3 isolates from different areas of the world.
